# *ZmASR3* from the Maize *ASR* Gene Family Positively Regulates Drought Tolerance in Transgenic Arabidopsis

**DOI:** 10.3390/ijms20092278

**Published:** 2019-05-08

**Authors:** Yani Liang, Yingli Jiang, Ming Du, Baoyan Li, Long Chen, Mingchao Chen, Demiao Jin, Jiandong Wu

**Affiliations:** 1National Engineering Laboratory of Crop Stress Resistance Breeding, School of Life Sciences, Anhui Agricultural University, Hefei 230036, China; 15955140258@163.com (Y.L.); yingli_jiang@163.com (Y.J.); mingdu@163.com (M.D.); cl16720019@163.com (L.C.); cmc1315538@gmail.com (M.C.); 15855281316@163.com (D.J.); 2Institute of Plant Protection, Yantai Academy of Agricultural Sciences, Yantai 265500, China; byli1314@163.com

**Keywords:** maize, *ASR*, drought tolerance, ROS, ABA

## Abstract

Abscisic acid (ABA)-, stress-, and ripening-induced (ASR) proteins are reported to be involved in drought stress responses. However, the function of maize *ASR* genes in enhancing drought tolerance is not known. Here, nine maize *ASR* members were cloned, and the molecular features of these genes were analyzed. Phenotype results of overexpression of maize *ZmASR3* gene in *Arabidopsis* showed lower malondialdehyde (MDA) levels and higher relative water content (RWC) and proline content than the wild type under drought conditions, demonstrating that *ZmASR3* can improve drought tolerance. Further experiments showed that *ZmASR3-*overexpressing transgenic lines displayed increased stomatal closure and reduced reactive oxygen species (ROS) accumulation by increasing the enzyme activities of superoxide dismutase (SOD) and catalase (CAT) under drought conditions. Moreover, overexpression of *ZmASR3* in *Arabidopsis* increased ABA content and reduced sensitivity to exogenous ABA in both the germination and post-germination stages. In addition, the ROS-related, stress-responsive, and ABA-dependent pathway genes were activated in transgenic lines under drought stress. Taken together, these results suggest that *ZmASR3* acts as a positive regulator of drought tolerance in plants.

## 1. Introduction

Plants are often exposed to many severe environmental conditions, among which drought is a major factor that adversely affects growth and development, subsequently reducing the yield of crops. To survive and adapt to these abiotic stresses, plant cells possess regulatory mechanisms and stress response pathways that allow the plant to respond rapidly to environmental changes [1]. An increasing number of studies have shown that plants can defend themselves against various stresses through synthesizing and activating a group of functional proteins, including transcription factors, enzymes, molecular chaperones, signaling molecules, antioxidants, and detoxification proteins [2,3,4,5]. For example, overexpression of *ZmHDZ4* is able to increase drought tolerance in transgenic rice [6]. The pepper ethylene-responsive transcription factor, *CaAIEF1*, confers tolerance to drought stress via an abscisic acid (ABA)-dependent pathway [7]. Although a series of stress-responsive genes have been identified, the functions of these stress-related genes are still not well characterized.

ABA is an important phytohormone that is involved in many physiological and biochemical processes in response to complex environmental stresses. Under drought stress, ABA accumulates in the leaf tissue and subsequently induces stomatal closure, as well as the formation of various protective metabolites and expression of stress-related genes [8]. In the ABA signaling pathway, ABA stimulates the generation of H_2_O_2_ by Nicotinamide Adenine Dinucleotide Phosphate (NADPH) oxidase in guard cells, and the generated H_2_O_2_ plays a vital role as an important signaling molecule in regulating stomatal closure by activating plasma membrane calcium channels [9]. For example, transgenic rice lines overexpressing *OsASR5* exhibited increasing endogenous ABA levels and stomatal closure compared with those of the wild type (WT) under drought stress conditions [7]. 

ABA-, stress-, and ripening-induced (ASR) proteins are part of a small protein family whose members are heat stable and highly hydrophilic [10]. *ASR* genes were first identified in tomatoes, and since then, many *ASR* genes have been reported in various species, including dicotyledonous, monocotyledonous, xylophyta, and herbaceous plants, whereas no *ASR* genes have been found in *Arabidopsis* [11,12,13,14,15,16,17]. ASR proteins have been reported to participate in the processes of plant development, senescence, and fruit ripening [18,19,20]. *ASR* genes have also been reported to respond to ABA and abiotic stresses. Overexpression of wheat *ASR* (*TaASR1*) in tobacco increased tolerance to drought by regulating the expression of stress/reactive oxygen species (ROS)-related genes [15]. A *Salicornia brachiata ASR* gene, *SbASR1*, increased tolerance to salt and drought stress by decreasing H_2_O_2_ and O_2_^.-^ content and regulating the expression of stress-related genes [21]. Although *ASR* genes were discovered and reported in response to abiotic stresses based on the transcriptomic analysis of maize leaves [14], little is known about the exact functions of maize *ASR* genes under abiotic stresses.

Maize (*Zea mays* L.) serves as an important food crop, whose production is adversely affected by abiotic stresses, especially drought conditions. Therefore, screening and characterization of the roles of stress-related genes from maize is essential to improve the resistance towards abiotic stresses. Here, we analyzed the *ASR* gene family from maize and investigated their function in transgenic lines under drought conditions. Overexpression of *ZmASR3* resulted in increased drought stress tolerance in plants by activating the antioxidant system and regulating the ABA-dependent pathway. Therefore, our results indicate that *ZmASR3* serves as an important target gene for enhancing the tolerance of drought stress in crop breeding programs.

## 2. Results

### 2.1. Identification and Sequence Analysis of ZmASR Proteins 

A number of *ASR* genes have been identified to respond to abiotic stresses [1,15,21], but the function of maize *ASR* genes in drought stress is still not known. To identify ASR proteins, the conserved ASR domain (PF02496) based on the hidden Markov model (HMM) from the Pfam database was used as a query in BLASTp searches of homologous sequences in the maize genome, and nine *ASR* members were identified. The exact information, including the open reading frame (ORF), molecular weight (MW), isoelectric point (pI), and chromosomal location of each *ASR* gene, is listed in Appendix A. Multiple sequence alignment revealed that these ZmASRs all harbor an N-terminal zinc-binding domain, a C-terminal nuclear targeting signal, and two abscisic acid/water deficit stress domains (ABA/WDS), similar to other known ASR protein family members (Figure 1). 

### 2.2. Expression of the ZmASR Genes in Different Tissues

Publicly available genome-wide transcript profiling was used to detect the expression patterns of *ZmASRs* from various tissues and organs of maize [22]. Most of the *ZmASRs* were ubiquitously expressed in all analyzed tissues, implying the function of *ZmASRs* in many development processes (Figure 2A). An exception to this was *ZmASR8*, which was rarely expressed in any tissue or organ. These results were also confirmed by quantitative real-time PCR (qRT-PCR) analysis of seven representative tissues of maize (Figure 2B). The *ZmASRs* have higher expression levels in roots and leaves compared with other tissues, except with respect to *ZmASR8*. 

### 2.3. Sub-Cellular Localization and Interaction Analysis of ZmASRs

To investigate the sub-cellular localization of ZmASRs, the ZmASR open reading frame (ORF) without the termination codon was fused upstream of the GFP reporter, under the control of the CaMV 35S promoter. The construct was introduced into leaves of *Nicotiana benthamiana* L. plants and observed under a confocal microscope. The results showed that ZmASR1, ZmASR2, ZmASR3, ZmASR4, ZmASR6, ZmASR7, ZmASR8, and ZmASR9 fusion proteins were only localized in the nucleus, whereas ZmASR5-GFP was localized in the cytosol (Figure 3). These results imply that ZmASR3 and other ASR proteins in the nucleus might act as transcription factors or function as chaperone-like proteins. 

A previous study reported that ASR proteins may act as molecular chaperones in plants [23]. We therefore hypothesized that ZmASRs serve as molecular chaperones in the formation of homodimers or heterodimers. To confirm this point, we generated pGBKT7-ZmASR and pGADT7-ZmASR constructs and performed yeast two-hybrid assays to analyze the interaction between these members. The results showed that only the yeast cells carrying pGBKT7-ZmASR3/pGADT7-ZmASR3 and pGBKT7-ZmASR4/pGADT7-ZmASR4 grew well on SD/-Ade/-His/-Leu/-Trp/X-α-Gal plates (Appendix A), suggesting that ZmASR3 and ZmASR4 may form a homodimer in plants.

### 2.4. Characterization of the Function of ZmASRs in Drought Tolerance 

Although a previous study based on transcriptomic analysis of maize leaves showed that *ZmASR1*, *ZmASR2*, *ZmASR4*, and *ZmASR7* were up-regulated under drought conditions, and *ZmASR1*, *ZmASR3*, and *ZmASR4* were up-regulated under ABA treatment [14], the function of *ZmASRs* in response to abiotic stresses in plants is still unclear. Therefore, we constructed a pCAMBIA1301 overexpression vector with *ZmASRs* and transformed these into *Arabidopsis* to analyze their function under conditions of mannitol stress and ABA treatment. Subsequently, three independent homozygous T3 lines (OE3-1, OE3-2, and OE3-3), carrying different constructs with different expression levels, were chosen for further studies (Appendix A). Phenotypic analysis showed that the transgenic lines overexpressing *ZmASR3* had longer roots than WT under mannitol treatment, whereas other *ZmASR-*overexpressing transgenic lines displayed no obvious differences (Appendix A). In addition, longer roots were also detected with ABA treatment of transgenic lines overexpressing *ZmASR2* and *ZmASR3* compared with WT. Based on this phenotype, the *ZmASR3*-overexpressing line was chosen for further studies in the following experiments.

### 2.5. Overexpression of ZmASR3 Enhances Drought Tolerance in Arabidopsis

To further evaluate the function of *ZmASR3* in drought tolerance, we performed a detailed analysis of *ZmASR3* transgenic lines and WT plants. We first examined the germination rates in transgenic plants and WT under 200 mM mannitol treatment. Almost 85.7% of *ZmASR3*-overexpressing seeds were able to germinate compared with only 70.2% of the WT seeds (Figure 4A,B). We then analyzed the root lengths and fresh weights under different concentrations of mannitol. Longer roots and increased fresh weights were detected in the *ZmASR3*-overexpressing lines compared with WT (Figure 4C–E). 

To analyze the phenotype of *ZmASR3* transgenic lines in soil under drought conditions, water was withheld for 15 days from two-week-old *ZmASR3*-overexpressing and WT seedlings, and then, the seedlings were transferred to normal conditions and grown for five days (Figure 5A). The survival rate of WT plants was only 38.5%, while an average of 79.4% of transgenic plants were recovered (Figure 5B). In addition, lower malondialdehyde (MDA) content and higher proline levels and relative water content were measured in *ZmASR3*-overexpressing lines compared with those of WT (Figure 5C–E). These results showed that overexpression of *ZmASR3* results in enhancing drought stress tolerance in *Arabidopsis*.

### 2.6. Overexpression of ZmASR3 Increases Stomatal Closure

Stomata play an important role in drought stress tolerance by regulating water and gas exchange in plants [24]. Therefore, we observed the stomatal closure of leaves in *ZmASR3*-overexpressing transgenic *Arabidopsis* and WT plants. As shown in Figure 6A,B, there were no apparent differences between completely open, partially open, and completely closed stomata in *ZmASR3*-overexpressing transgenic lines and WT plants under normal conditions. However, an average of 53.6% stomata completely closed in three different transgenic lines, while 22.1% completely closed in WT plants under drought conditions. In contrast, an average of 37.3% partially open and 9.1% completely open stomata were observed in transgenic lines compared with WT plants, which showed 58.3% partially open and 19.6% completely open stomata. These results show that overexpression of *ZmASR3* increases stomatal closure.

### 2.7. Overexpression of ZmASR3 Increases Drought Tolerance by Decreasing H_2_O_2_ Accumulation and Increasing Superoxide Dismutase (SOD) and Catalase (CAT) Activities

In a previous study, it was shown that H_2_O_2_ accumulation can lead to stomatal closure [7]. Therefore, it was necessary to examine ROS levels in the *ZmASR3*-overexpressing lines and WT. After 15 days under drought conditions, leaf samples were collected and stained with diaminobenzidine tetrahydrochloride (DAB) solution. There was no obvious difference between transgenic *Arabidopsis* and WT plants under normal conditions. In contrast, under drought stress, leaves of WT plants exhibited stronger staining than those of transgenic lines (Figure 7A). These results show that WT leaves accumulated more ROS compared with leaves of transgenic lines under drought stress. 

Enzymatic antioxidants play important roles in ROS scavenging. Therefore, the activities of antioxidant enzymes were measured in WT and transgenic plants. The activities of SOD and CAT were not apparently different between transgenic and WT plants under normal conditions. Under drought conditions, the three transgenic lines all exhibited obviously higher SOD and CAT activities compared with those of WT plants (Figure 7B,C). However, no obvious difference in peroxidase (POD) activities was observed between transgenic and WT plants under drought conditions (Appendix A). These results showed that overexpression of *ZmASR3* reduced ROS accumulation by increasing the activities of SOD and CAT enzymes under drought conditions.

### 2.8. ZmASR3 Regulates the Expression of ROS-Related and Stress-Responsive Genes under Drought Conditions 

The mRNA expression levels of stress-related genes were also analyzed to confirm the function of *ZmASR3* in drought tolerance. Selected genes for this analysis included *AtCAT3* and *AtSOD1* involved in ROS detoxification, and *AtLTP3, AtSOS1*, and *AtRD29B* related to stress defense. The results showed that the expression of all tested genes was up-regulated in the *ZmASR3-*overexpressing transgenic line under drought conditions (Figure 8), suggesting that *ZmASR3* can affect the expression of ROS-related and stress-responsive genes under drought conditions.

### 2.9. Overexpression of ZmASR3 Increased Endogenous ABA Levels and Reduced Sensitivity to Exogenous ABA

ABA can result in stomatal closure and enhance drought tolerance, so we measured the endogenous ABA levels in *ZmASR3*-overexpressing lines and WT. As shown in Figure 9, the ABA levels were clearly higher in WT and transgenic lines after they were subjected to drought conditions compared with the controls. In addition, ABA content was higher in *ZmASR3*-overexpressing lines than that in WT plants after they were subjected to drought conditions. However, there was no apparent difference between ABA levels in WT and transgenic lines under normal conditions. These results suggest that *ZmASR3* may play an important role in ABA signaling. To confirm this speculation, we investigated the ABA sensitivity of *ZmASR3*-overexpressing lines in the germination and post-germination stages. As shown in Appendix A, no obvious difference in germination rates, root lengths, and lateral root numbers were observed between *ZmASR3*-overexpressing lines and WT under normal conditions. However, higher germination rates, longer root lengths, and an increase in lateral root numbers were detected in *ZmASR3*-overexpressing lines compared with WT lines following ABA treatment.

### 2.10. Overexpression of ZmASR3 Enhances Drought Tolerance via an ABA-Dependent Pathway

In order to explore whether *ZmASR3* functions in the ABA signaling pathway under mannitol stress, tungstate (Tu), a well-known inhibitor of ABA biosynthesis, was applied [25]. The results show that root lengths were noticeably reduced in the transgenic lines and WT in the medium with Tu treatment under normal conditions, with no obvious difference between transgenic lines and WT (Appendix A). Under drought conditions, a higher percentage of leaf opening and greening and longer roots were detected in *ZmASR3-*overexpressing lines compared with WT plants in the medium with Tu treatment (Figure 10). These results indicate that *ZmASR3* enhances the tolerance to drought probably via an ABA pathway. To further confirm this hypothesis, the transcript levels of well-known marker genes in the ABA-dependent pathway (*NCED3* and *SnRK2.6*) and ABA-independent pathway (*COR15* and *DREB2A*) [15,26,27,28,29,30,31] were analyzed in *ZmASR3*-overexpressing lines under different treatments. As shown in Figure 11, *ZmASR3* expression was 10.4-fold and 3.6-fold higher after mannitol and ABA treatments, respectively, while treatment with Tu reduced the fold increase to 2.3-fold at 12 h after mannitol treatment. No obvious difference in *ZmASR3* expression was detected between treatment with Tu and the controls. Additionally, the marker genes *NCED3* and *SnRK2.6* in the ABA-dependent pathway, presented a similar trend with *ZmASR3* expression after different treatments. However, Tu treatment showed no noticeable reduction in the expression of *COR15* and *DREB2A* in the ABA-independent pathway under mannitol treatment. 

## 3. Discussion 

Since the first discovery of tomato ASR1, an increasing number of ASR proteins have been reported in different species. ASR proteins play important roles in plant development, responding to abiotic stresses, and fruit ripening [32]. Maize, which serves as an important crop for humans and livestock, has been less studied than other crops. Although maize *ASR* genes can be induced by different stress treatments based on transcriptomic analysis [14], little is known about the detailed function of *ZmASRs* under abiotic stress. In this study, we identified nine *ASR* genes and characterized the function of ASR proteins in response to drought. 

Multiple sequence alignment results show that all the ZmASR proteins contained an N-terminal zinc-binding domain and a C-terminal nuclear targeting signal, which was consistent with a previous report [33]. Interestingly, sub-cellular localization studies showed that ZmASR5 localized to the cytosol, whereas other ZmASRs only localized to the nucleus. These results implied that ZmASR3 and other ASR proteins in the nucleus might act as transcription factors or function as chaperone-like proteins. In fact, this has also been speculated by others [32,34]. For example, *SbASR-1* can enhance salinity and drought endurance in transgenic groundnuts and act as a transcription factor [21]. In addition, ASR proteins act as molecular chaperones in plants [23]. To confirm this point, we also detected an interaction between these proteins and found that ZmASR3 and ZmASR4 could form a homodimer in yeast, suggesting that they may serve as molecular chaperones in plants. More experiments would be needed to confirm this theory in the future. 

ASR proteins are involved in abiotic stress responses in many plants [14,15,35]. Here, we obtained transgenic *Arabidopsis* lines overexpressing nine maize *ASR* genes and analyzed their root lengths under mannitol and ABA treatment. Apparent differences in root lengths were observed only in *ZmASR2*- and *ZmASR3*-overexpressing transgenic lines under ABA treatment. However, there was a noticeable difference in root length only in the *ZmASR3*-overexpressing transgenic line under mannitol treatment. Therefore, *ZmASR3* was selected for further studies. Overexpression of *ZmASR3* in *Arabidopsis* obviously increased drought stress tolerance, as demonstrated by transgenic lines exhibiting better growth performance and higher survival rates compared with WT under drought conditions. These data indicate that *ZmASR3* may play an important role in abiotic stress tolerance in maize. Additionally, *ZmASR3-*overexpressing transgenic lines displaying various physiological traits may explain the mechanisms underpinning abiotic stress tolerance. Firstly, MDA and proline are two important indicators in responses to plant stress [36,37,38,39]. In agreement with this, *ZmASR3*-overexpressing transgenic lines had lower MDA content and higher proline content compared with WT under drought conditions. Secondly, in general, plants can maintain the balance of ROS in cells. However, plants can overproduce ROS, which leads to cell damage by oxidizing DNA and proteins under abiotic stresses [38,40,41]. In addition, oxidative damage can be minimized by increasing the activities of CAT, POD, and SOD enzymes under drought and salt conditions [42,43]. For example, *TaASR1-*overexpressing transgenic tobacco lines reduced the accumulation of ROS [15]. In this study, high CAT and SOD enzyme activities were also detected in transgenic lines under drought conditions, which was helpful for maintaining low ROS levels. Lastly, ROS-related and stress-responsive genes were activated in transgenic lines. 

ABA serves as a key phytohormone that is involved in many biology processes in plants, including plant growth and development, regulation of stomatal movements and abiotic stress responses [44]. Previous studies have reported that *WRKY57* enhanced drought stress tolerance by accumulating ABA, which limits transpirational water loss by regulating the stomatal apertures in *Arabidopsis* [45]. In agreement with this, higher ABA content was also present in *ZmASR3*-overexpressing transgenic lines. This leads to the closure of stomata to limit transpirational water loss, which thereby improves drought stress tolerance. In addition, plants confer abiotic stress tolerance mainly through the ABA-dependent and ABA-independent pathways [46]. In this study, the marker genes of the ABA-dependent pathway (*NCED3* and *SnRK2.6*), were activated in *ZmASR3-*overexpressing transgenic lines under drought stress. However, the expression of *NCED3* and *SnRK2.6* genes were reduced with Tu treatment. These results suggest that *ZmASR3* enhances drought tolerance via an ABA-dependent pathway. Surprisingly, an increased resistance to exogenous ABA in transgenic plants (Appendix A) was also detected. In fact, similar to *ZmASR3-*overexpressing plants, *abi1-1* is ABA-insensitive at the germination stage and has higher ABA content, but it also has a high leaf transpiration rate and low proline content [47]. In addition, the constitutive expression of *OsABIL2* or *OsABIL2^G183D^* in *Arabidopsis* or rice decreased ABA sensitivity to differing degrees. Moreover, the transgenic rice expressing *OsABIL2^G183D^* exhibited improved seedling growth under low temperature [48]. Our results could suggest that *ZmASR3* uses an ABA signaling pathway that probably does not require ABI1-like PP2Cs.

In conclusion, our study has shown that overexpression of *ZmASR3* confers drought stress tolerance in *Arabidopsis*. *ZmASR3* conferred drought tolerance by regulating the expression of antioxidant and drought-related genes and enhancing antioxidant enzyme activity. Furthermore, *ZmASR3* used the ABA signaling pathway by accumulating ABA to regulate stomatal closure under drought conditions and regulating the expression of ABA/stress-responsive genes. 

## 4. Methods and Materials

### 4.1. Identification and Multiple Sequence Alignment of ASR Proteins

To identify ASR proteins, the conserved ASR domain (PF02496) was used in BLASTp queries for searching maize homologous ASR proteins. All candidate protein sequences of genes were verified using Pfam software (PF00642; http://pfam.sanger.ac.uk/) and SMART tool databases (http://smart.embl-heidelberg.de/), based on the conserved ABA/WDS domain. The online ExPASy program (http://www.expasy.org/tools/) was used to analyze the features of ASR proteins, including the molecular weight (kDa), isoelectric point (pI), and protein length. Different sequences of ZmASR proteins were characterized based on ClustalX software (vl.83, Invitrogen, Carlsbad, CA, USA).

### 4.2. Plant Growing Conditions and Stress Treatments 

All plant materials (maize, *Arabidopsis thaliana*, *Nicotiana benthamiana*) were grown for different experiments as described previously [49]. Different maize tissues, including roots, stems, leaves, tassels, cornsilk, embryo, and stegophyll were collected for tissue expression profile analysis. For drought stress treatments, sterilized seeds of T3 *Arabidopsis* were grown on Murashige and Skoog (MS) solid medium at 4 °C for three days and then placed in MS agar plates supplemented with different concentrations of mannitol and cultured vertically for 10 days. For drought conditions, two-week-old seedlings grown in soil were exposed to drought stress by withholding water for 15 days and then re-watered for five days.

### 4.3. Constructs and Plant Transformation

Gene-specific primers (Appendix A) were designed to clone *ZmASR* using the PCR-based infusion clone system (GBclonart, Takara, Tokyo, Japan). The PCR products were inserted into different vectors for subsequent experiments. 

Wild-type plants (Col-0) were transformed using the floral dipping method to acquire *35S*::*ZmASR* transgenic lines. Seeds from each T1 plant were individually collected and selected on MS medium containing 25 μg mL^−1^ hygromycin. 

### 4.4. Sub-Cellular Localization Assay 

Full-length cDNAs of *ZmASR*s were amplified and then inserted into a pCAMBIAI1305 vector driven by a Cauliflower Mosaic Virus 35S (CaMV35S) promoter. The *35S*::*ZmASR-GFP* and *GFP* control vector were transformed into *Agrobacterium tumefaciens* (strain GV3101) cells. *Agrobacteria* carrying the constructs were infiltrated into the leaves of five- to six-week-old *Nicotiana benthamiana* plants. After 48 h, the infected leaves were observed using a Zeiss Microsystems LSM710 microscope (Jena, Germany). 

### 4.5. Yeast Two-Hybrid Assays

For analyzing the interaction between ZmASRs, pGADT7-ZmASR and pGBKT7-ZmASR fusion vectors were generated and transformed into yeast strain AH109, according to the manufacturer’s instructions (Clontech, Mountain View, CA, USA). The fusion vectors were transformed into AH109 yeast competent cells and were then grown on culture medium (SD/-Trp/-Leu, SD/-Trp/-Leu/-Ade/-His) at 30 °C constant temperature.

### 4.6. Total RNA Isolation and qRT-PCR analysis

Total RNA from both *Arabidopsis* and maize (B73) tissues was extracted by the RNAiso plus (TaKaRa, Dalian, China) method. The cDNA synthesis reaction was performed with SuperScript^TM^ III reverse transcriptase (Invitrogen). qRT-PCR was performed on each cDNA template using SYBR Green Master Mix on an ABI 7300 Real-Time system, according to the manufacturer’s protocol (Applied Biosystems). Each 20 μL of mixture sample was used for the following PCR cycles: 95 °C for 10 s and 60 °C for 60 s. The data were analyzed by 2^−ΔΔ*C*t^ method. The primers are listed in Appendix A. Each qRT-PCR was run in quadruplicate.

### 4.7. Sensitivity Analysis of Exogenous ABA

Plants of WT and transgenic seeds were grown on MS medium plates containing 0, 0.3, 20, and 40 µM ABA solutions. Subsequently, the germination rates, root lengths, and the number of lateral roots of transgenic and WT seedlings were analyzed.

Measurement of the stomatal aperture was carried out as described previously [50]. Leaves of WT and overexpressing *ZmASR3* plants were soaked in the buffer solution: CaCl_2_ (0.2 mM), KCl (10 mM), MES (10 mM), and KOH (pH 6.15). After soaking in the buffer for 2 h in the light, the samples were transferred into 10 µM ABA solution immediately for 2 h. Then, samples were observed under the microscope.

### 4.8. DAB Staining, Determination of Physiological and Biochemical Activities, and ABA Content

Rosette leaves of transgenic *Arabidopsis* and wild-type plants were collected for detecting H_2_O_2_ accumulation using DAB. The leaves were soaked in DAB solution (1 mg mL^−1^ DAB in 10 mM sodium phosphate buffer) and were treated for 24 h with gentle shaking in the darkness. Leaves were washed with absolute ethyl alcohol solution, transferred to 100 °C for 10 min, and photographed.

We measured the CAT, POD and SOD activities by spectrophotometric analysis: about 0.5 g seedlings were extracted with 5 mL of buffer solution containing 50 µmol L^−1^ phosphate. The activity of CAT and SOD was measured following previous methods [51,52]. POD activity was measured as the absorbance at 470 nm [53]. To determine ABA content, 1 g seedlings in the four-leaf stage were homogenized in 30 mL of buffer solution (isopropyl alcohol–hydrochloric acid/dichloromethane, 1:2). A HPLC-MS/MS method was used to measure the ABA content [54].

### 4.9. Measurement of RWC and Proline and MDA Contents

RWC was measured by the method previously described [6]. RWC was calculated according to the equation: RWC (%) = [(FW − DW)/(TW − DW)] × 100; FW and TW represent the fresh weight and turgid weight (leaves were soaked in water), respectively; DW represents the dry weight (leaves were treated at 80 °C for 24 h). Proline and MDA contents were measured by the methods described previously [55,56,57,58].

### 4.10. Statistical Analysis

The data are expressed as the mean values ± standard errors (S.E.) of at least three replicates and three biological repeats. Two-factor multivariate analysis of variance was used to test for the effects of Tu, ABA, and mannitol on gene expression. Differences between means were tested using LSD tests at a 0.05 significance level. All statistical tests were performed using the DPSv7.05 software to compare the means of different treatments (*n* ≥ 3).

## Figures and Tables

**Figure 1 ijms-20-02278-f001:**
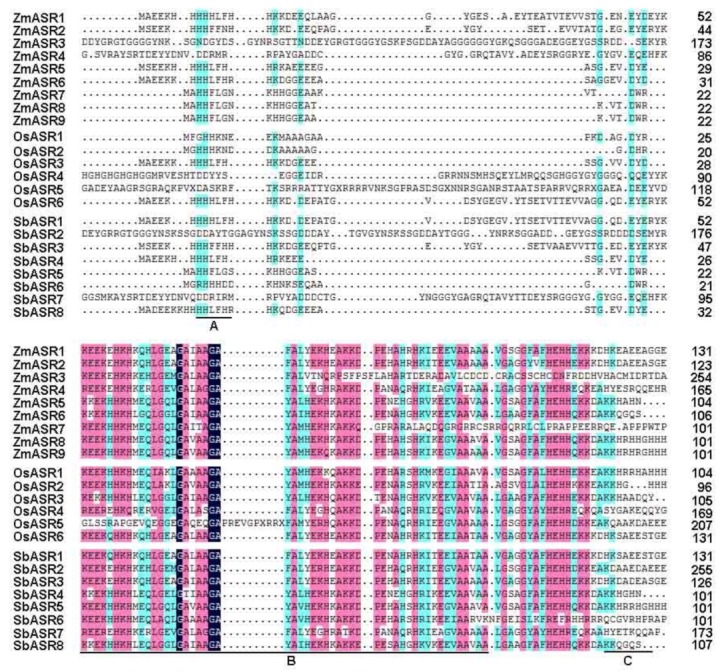
Alignment of amino acid sequences of abscisic acid (ABA)-, stress-, and ripening-induced (ASR) proteins. (**A**) The zinc-binding region; (**B**) the ABA/water deficit stress (WDS) domain; and (**C**) the putative nuclear targeting signal. The shading of the alignment presents identical residues in magenta, blue and dark blue colors, and the high conserved amino are marked at the bottom.

**Figure 2 ijms-20-02278-f002:**
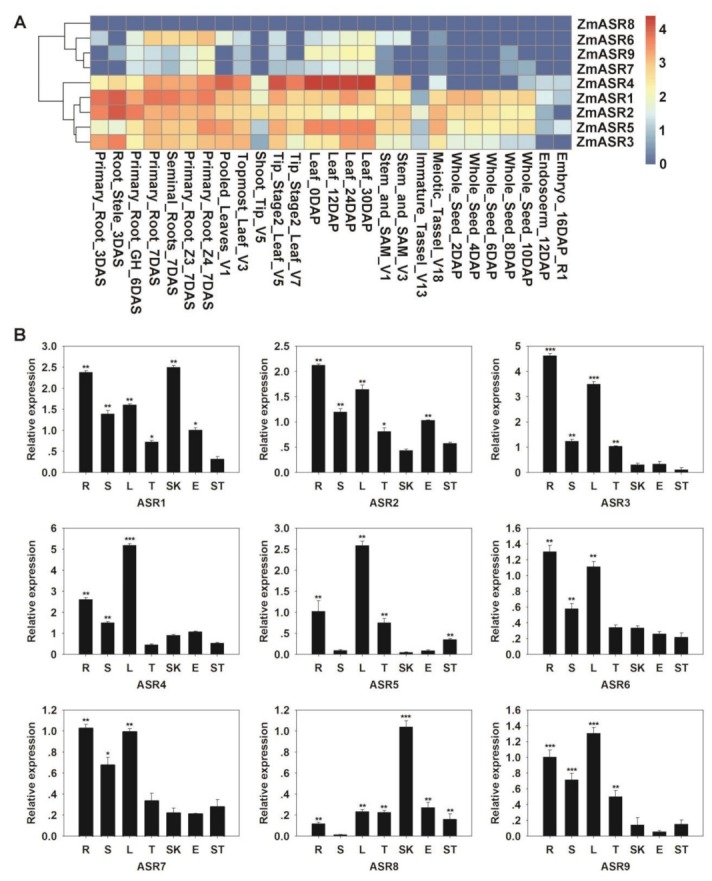
Expression pattern analysis of *ZmASR* genes. (**A**) Expression patterns of *ZmASRs* in different tissues. Red, yellow, and white indicate high, medium, and low levels of gene expression, respectively. (**B**) Tissue-specific expression patterns of the *ZmASR* genes. R (root); S (stem); L (leaf); T (tassel); SK (cornsilk); E (embryo); and ST (stegophyll). Vertical bars indicate means ± S.E. Significant differences: * *p* < 0.05; ** *p* < 0.01; *** *p* < 0.001.

**Figure 3 ijms-20-02278-f003:**
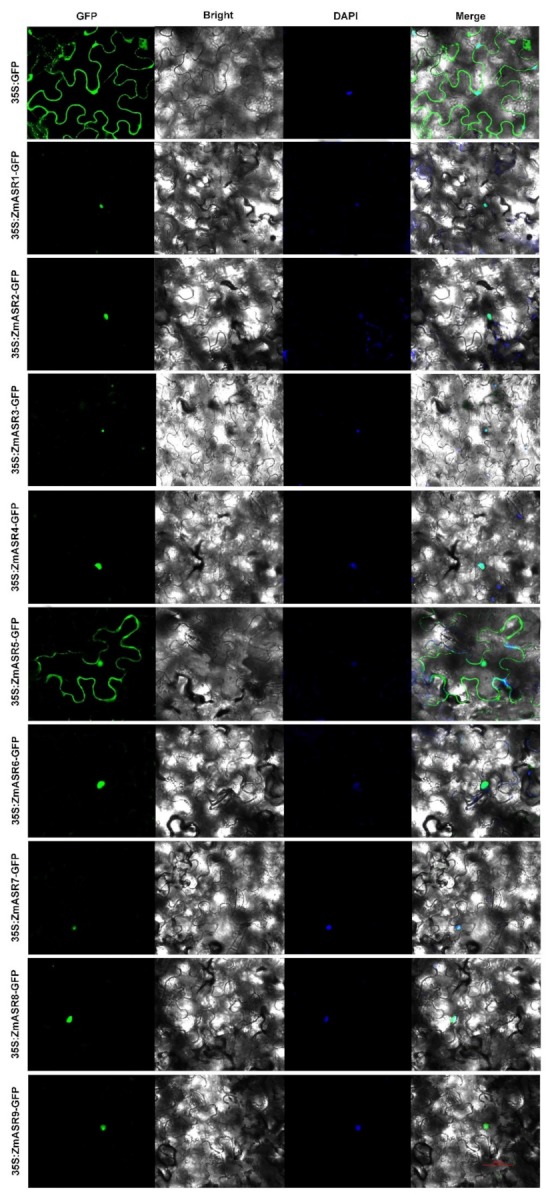
Sub-cellular localization analysis of ZmASRs. Sub-cellular localization of the ZmASR-GFP constructs in tobacco leaf epidermal cells. Green color is GFP protein signal, and blue color represents DAPI stained for nucleus. Scale bar = 50 µm.

**Figure 4 ijms-20-02278-f004:**
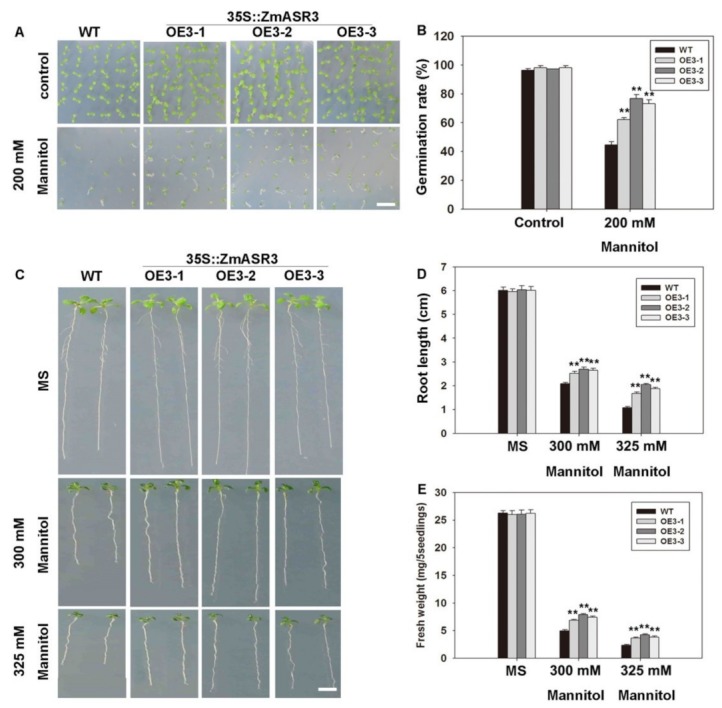
Overexpression of *ZmASR3* enhances drought tolerance in the germination and post-germination stages in *Arabidopsis*. (**A**) Phenotype and (**B**) quantitative evaluation of germination rates of the wild-type (WT) and three transgenic lines. Photographs were taken eight days after germination. (**C**) Photographs and measurements of (**D**) root lengths and (**E**) fresh weights of WT and transgenic seedlings in the post-germination stage under control (Murashige and Skoog (MS)) and osmotic stress (mannitol) conditions. All samples were measured in three independent experiments (each with 100 seeds for each line). Vertical bars indicate means ± S.E. (*n* ≥ 3). Significant differences: ** *p* < 0.01. Scale bar = 1 cm.

**Figure 5 ijms-20-02278-f005:**
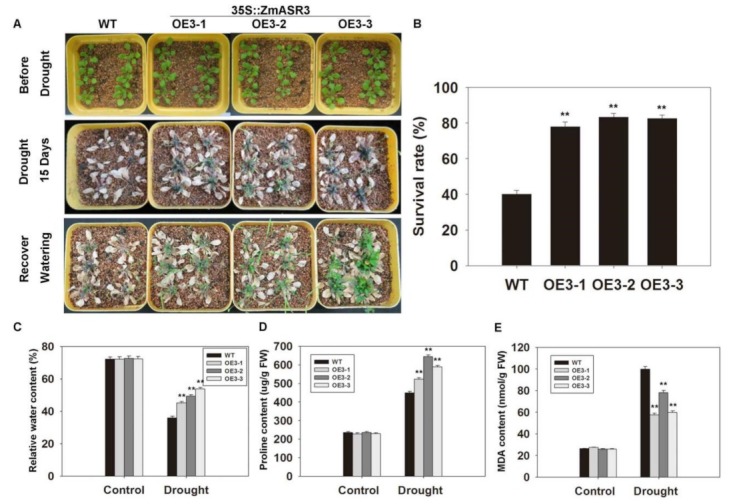
Overexpression of *ZmASR3* enhances drought tolerance in *Arabidopsis* in soil. (**A**) Drought tolerance of three transgenic lines and wild-type plants. (**B**) Survival rate, (**C**) relative water content, (**D**) proline content, and (**E**) MDA content were subsequently measured. All samples were measured in three independent experiments (each with 50 seedlings for each line). In (**B**–**E**), vertical bars indicate means ± S.E. (*n* ≥ 3). Significant differences: ** *p* < 0.01.

**Figure 6 ijms-20-02278-f006:**
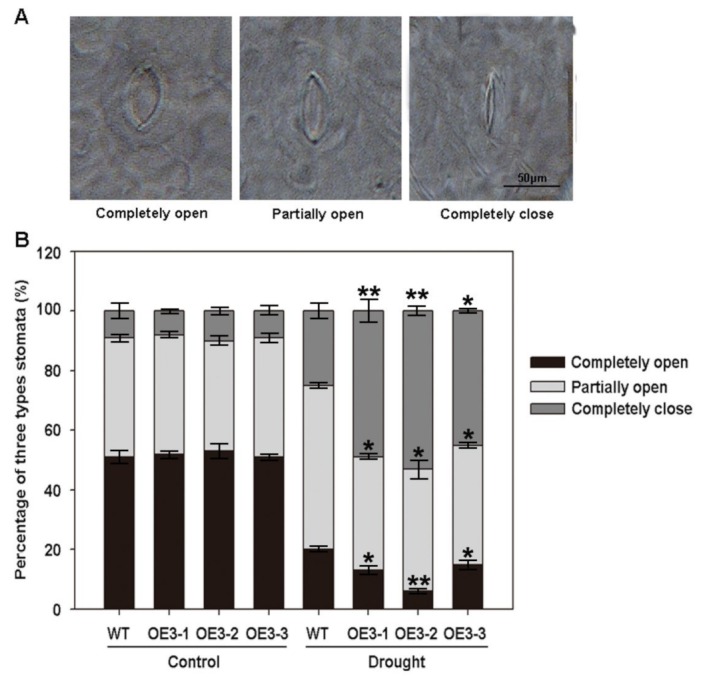
Overexpression of *ZmASR3* increases stomatal closure. (**A**) Observation of stomatal closure in transgenic *Arabidopsis* and WT plant leaves. Scale bar = 50 μm. (**B**) The percentage of completely open, partially open, and completely closed stomata were analyzed in the leaves of *ZmASR3* transgenic *Arabidopsis* and WT plants under normal conditions and drought stress (200 stomata were analyzed for each line). Vertical bars indicate means ± S.E. (*n* ≥ 3). Significant differences: * *p* < 0.05; ** *p* < 0.01. Scale bar = 50 μm.

**Figure 7 ijms-20-02278-f007:**
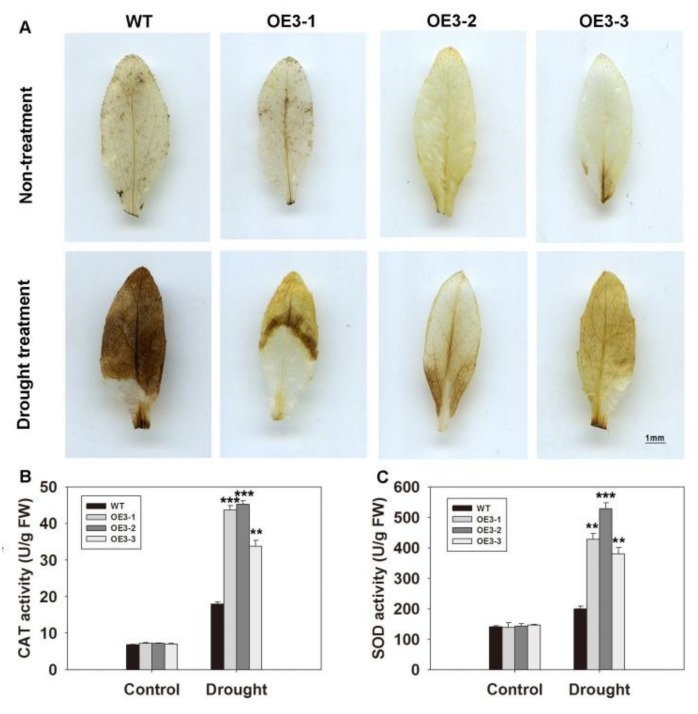
Overexpression of *ZmASR3* increases drought tolerance by decreasing H_2_O_2_ accumulation and increasing SOD and CAT activities. (**A**) H_2_O_2_ accumulation in the three transgenic lines and wild-type plants. Two-week-old *Arabidopsis* plants had water withheld for 15 days and were analyzed. Scale bar = 1 mm. (**B**) Analysis of CAT activity. (**C**) Analysis of SOD activity. All samples were measured in three independent experiments. For (**B**,**C**), vertical bars indicate means ± S.E. (*n* ≥ 3). Significant differences: ** *p* < 0.01; *** *p* < 0.001.

**Figure 8 ijms-20-02278-f008:**
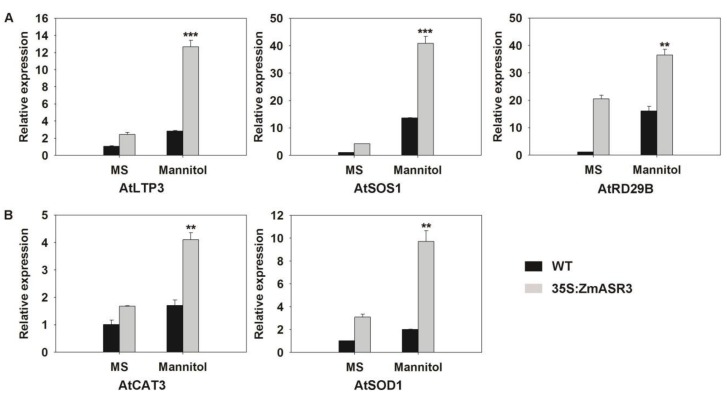
Analysis of expression levels of (**A**) stress-responsive and (**B**) reactive oxygen species (ROS)-related genes in the WT and transgenic line under control and osmotic stress conditions. The experiment was repeated three times with three biological repeats. Vertical bars indicate means ± S.E. (*n* ≥ 3). Significant differences: ** *p* < 0.01; *** *p* < 0.001.

**Figure 9 ijms-20-02278-f009:**
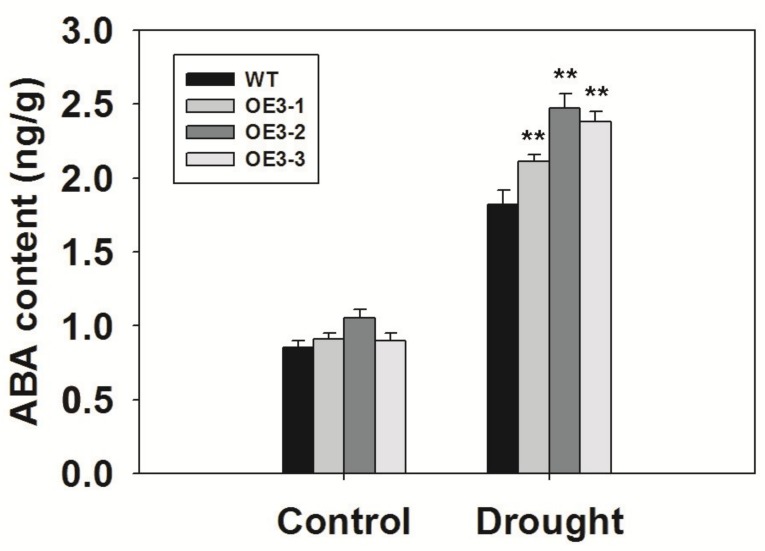
Overexpression of *ZmASR3* increases the ABA content under drought treatment. The experiment was repeated three times with three biological replicates. Vertical bars indicate means ± S.E. (*n* ≥ 3). Significant differences: ** *p* < 0.01.

**Figure 10 ijms-20-02278-f010:**
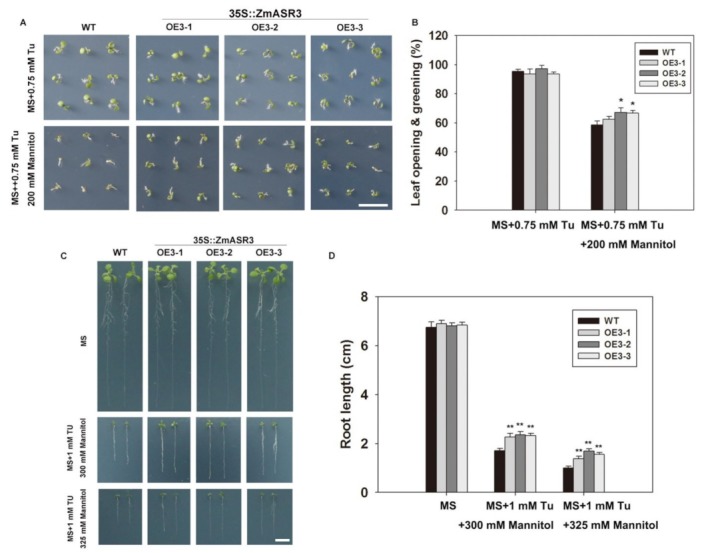
ABA signaling is involved in induction of *ZmASR3* with mannitol treatment. (**A**) Analysis of the germination of the three transgenic lines and wild-type plants. Seeds were germinated and grown for eight days on half MS medium containing 0.75 mM Tu and 0.75 mM Tu + 200 mM mannitol. (**B**) Germination rates of seeds shown in (**A**), as counted by leaf opening and greening. All samples were measured in triplicate. (**C**) Three-day-old seedlings grown on MS medium were transferred to fresh medium containing 1 mM Tu + 300 mM mannitol and 1 mM Tu + 325 mM mannitol for seven days. (**D**) Root lengths of the seedlings shown in (**C**). For (**B**,**D**), Scale bar = 1 cm vertical bars indicate means ± S.E. (*n* ≥ 3). Significant differences: * *p* < 0.05; ** *p* < 0.01.

**Figure 11 ijms-20-02278-f011:**
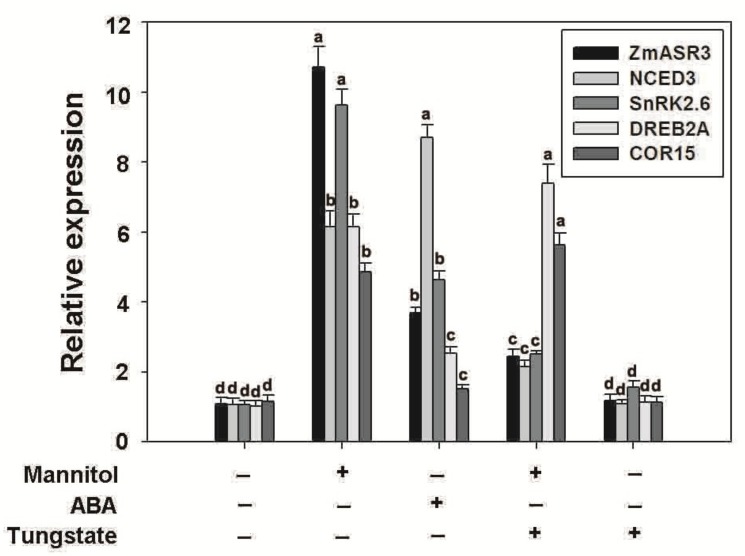
Effects of Tu and mannitol on the expression of stress/ABA-related genes in *ZmASR3*-overexpressing transgenic lines and WT. Marker genes in the ABA-dependent pathway (*NCED3* and *SnRK2.6*) and ABA-independent pathway (*COR15* and *DREB2A*) were analyzed in *ZmASR3*-overexpressing lines under different treatments. Vertical bars indicate means ± S.E. (*n* ≥ 3). ANOVA was used to reveal significant differences. Different letters above the column indicate statistical significance at the *p* = 0.05 level. + presents “treatment”, − presents “no treatment”.

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
