# Peer review of "ZmASR3* from the Maize *ASR* Gene Family Positively Regulates Drought Tolerance in Transgenic Arabidopsis"

_ijms, 2019, doi:10.3390/ijms20092278_

Round 1
Reviewer 1 Report
This paper reports the characterization of the ASR gene family in maize (nine genes), mainly based on the phenotyping of ASR-overexpressing plants in Arabidopsis. Since ASR genes have been reported to be involved in drought resistance in tobacco, it was relevant to look for the function of these gene in maize, a drought-sensitive major crop species. The authors identified the ASR genes in maize, reported their transcription pattern from publicly available data, localized the proteins in tobacco cells, and phenotyped over-expressing lines on mannitol. They isolated ASR3 as a gene involved in drought resistance, and addressed the different aspects and the mechanism of this resistance. ZmASR3 seems to have a broad effect on stress response.
There is a big amount of data showing the phenotype of ASR-overexpressing plants. Most of them depicting the resistance phenotype of ZmASR3 and the role of ROS, provided with statistical analyses, are convincing. But I have questions regarding the tungstate treatment. Tungstate is a competitor of molybdate and thus not only affects aldehyde oxidase in ABA synthesis but also other aldehyde oxidases, nitrate reductase, xanthine dehydrogenase, and it is also a heavy metal. The tungstate treatment alone has a drastic effect on root length, apparently stronger than the mannitol + tungsten treatment (if we compare fig. 6D and S6), which is surprising. It would have been better to cross the overexpressing lines with a mutant line defective in an enzyme of ABA synthesis. The other questionable point is the apparent contradiction between the increased resistance to exogenous ABA of overexpressing plants (fig. S5) and their resistance to drought stress and increased stomatal closure under drought, if drought resistance is mediated by ABA. This should at least be discussed.
Other (minor) concerns:
- How the dimerization of ASRs can support their role as chaperones?
- ZmASR5 seems to be in the cytosol rather than at the plasma membrane (figure 3).
- Many sentences are very difficult to read, with problems of syntax…
Author Response
Thank you for your comments concerning our manuscript. The comments are all valuable and very helpful for revising and improving our paper. In addition, they will also provide important guidance for our future research. We have studied the comments carefully and have made corrections accordingly to the new manuscript which we hope will meet with your approval. The main corrections in the marked versions of the revised manuscript and the responses to the reviewers’ comments are as follows.
Point 1: There is a big amount of data showing the phenotype of ASR-overexpressing plants. Most of them depicting the resistance phenotype of ZmASR3 and the role of ROS, provided with statistical analyses, are convincing. But I have questions regarding the tungstate treatment. Tungstate is a competitor of molybdate and thus not only affects aldehyde oxidase in ABA synthesis but also other aldehyde oxidases, nitrate reductase, xanthine dehydrogenase, and it is also a heavy metal. The tungstate treatment alone has a drastic effect on root length, apparently stronger than the mannitol + tungsten treatment (if we compare fig. 6D and S6), which is surprising.
Response 1: Thank you for your professional suggestions. Tungstate (Tu) as an well-known inhibitor of ABA biosynthesis was applied in many published papers; examples are Aying et al., 2011 and Cai et al., 2017. About the surprising results between fig. 10D (not fig. 6D) and S6, it may be partially explained by our results. First, the significantly difference is mainly present in transgenic lines, Secondly, transgenic lines had higher ABA content than wild type under drought stress (fig. 9). So transgenic lines had longer roots under the mannitol + tungsten treatment.
Aying, Z.; Jun, Z.; Jianhua, Z.; Nenghui, Y.; Hong, Z.; Mingpu, T.; Mingyi, J. Nitric oxide mediates brassinosteroid-induced ABA biosynthesis involved in oxidative stress tolerance in maize leaves. Plant & Cell Physiology 2011, 52, 181.
Cai, R.; Dai, W.; Zhang, C.; Wang, Y.; Wu, M.; Zhao, Y.; Ma, Q.; Xiang, Y.; Cheng, B. The maize WRKY transcription factor ZmWRKY17 negatively regulates salt stress tolerance in transgenic Arabidopsis plants. Planta 2017, 246, 1215-1231.
Point 2: It would have been better to cross the overexpressing lines with a mutant line defective in an enzyme of ABA synthesis. The other questionable point is the apparent contradiction between the increased resistance to exogenous ABA of overexpressing plants (fig. S5) and their resistance to drought stress and increased stomatal closure under drought, if drought resistance is mediated by ABA. This should at least be discussed.
Response 2: The comments are valuable and very helpful for revising and improving our paper. In addition, it will also provide important guidance for our future research. However, the cross experiment may be not completed in this study. About the sensitivity analysis of exogenous ABA, we also noticed the apparent contradiction. However, all experiments were performed with three biological and three technical replicates. The related discussion was also included in the new manuscript. Thank you again!
Point 3: Other (minor) concerns:
- How the dimerization of ASRs can support their role as chaperones?
- ZmASR5 seems to be in the cytosol rather than at the plasma membrane (figure 3).
- Many sentences are very difficult to read, with problems of syntax…
Response 3: Thank you for your comments.
The dimerization of ASRs can not support their role as chaperones in plants. In the text, we only stated that it is a possible speculation, but more experiments would be needed to demonstrate it.
Thank you for your warning, we modified it in the new manuscript.
We carefully checked the grammar and syntax of the manuscript which we hope will meet with your approval.
Reviewer 2 Report
· Please shorten the title to make easier understanding for all readers.
· Line 13: “However, nothing is known about the ASR function in maize.” How did the authors think there is nothings about that, while I listed some of published papers about ASR in maize?
1. Bukan M;Sarcevic H;Buhinicek I;Palaversic B;Lewis RS, Kozumplik V. Stalk Rot Resistance in Maksimir 3 Synthetic Maize Population after Four Cycles of Recurrent Selection. Genetika-Belgrade. 2013,45(3),921-928, 10.2298/Gensr1303921b.
2. Cota LV;da Costa RV;Silva DD;Casela CR, Parreira DF. Quantification of Yield Losses Due to Anthracnose Stalk Rot on Corn in Brazilian Conditions. Journal of Phytopathology. 2012,160(11-12),680-684, 10.1111/jph.12008.
4. Li XY;Li LJ;Zuo SY;Li J, Wei S. Differentially expressed ZmASR genes associated with chilling tolerance in maize (Zea mays) varieties. Functional Plant Biology. 2018,45(12),1173-1180, 10.1071/FP17356.
5. Virlouvet L;Jacquemot MP;Gerentes D;Corti H;Bouton S;Gilard F;Valot B;Trouverie J;Tcherkez G;Falque M et al. The ZmASR1 Protein Influences Branched-Chain Amino Acid Biosynthesis and Maintains Kernel Yield in Maize under Water-Limited Conditions. Plant Physiology. 2011,157(2),917-936, 10.1104/pp.111.176818.
6. Zhang J;Zhu QS;Yu HJ;Li L;Zhang GQ;Chen X;Jiang MY, Tan M. Comprehensive Analysis of the Cadmium Tolerance of Abscisic Acid-, Stress- and Ripening-Induced Proteins (ASRs) in Maize. International Journal of Molecular Sciences. 2019,20(1), Artn 133. 10.3390/Ijms20010133.
7. Zhu JW;Huang XL;Liu T;Gao SG, Chen J. Cloning and function analysis of a drought-inducible gene associated with resistance to Curvularia leaf spot in maize. Molecular Biology Reports. 2012,39(8),7919-7926, 10.1007/s11033-012-1636-6.
· Lines 64 and 65: “abiotic stress” should be abiotic stresses and also in across all the manuscript.
· Line 73: “but the exact function of maize ASR genes were still not well known”, what does it mean? Please explain clearly.
· Line 76: “member” change to members.
· Lines 66 and 74: ASR gene should be in italic and also across all the manuscript.
· Figure 2 caption: The authors have to analyze transcript levels to determine significant differences and show it using asterisk. Also which type of bars have been presented?
· Line 117: “the function of ZmASRs in responses to the abiotic stress still remain unclear in plants” please check for grammar.
· Line 120: The authors how did detected the homozygous lines? In biological processes sometimes may generate heterozygous embryos from homozygous and also reversely accidently.
· Supplemental figure 2: When the authors are talking about relative expression, they should analyze data via comparison statistical analyses and one control to exhibit significant differences between data. This process also has to be for supplemental figure 3. In addition, the authors have Witten SE, but according to the statistical analyses it should be Standard deviation. Moreover, I cannot find any asterisk in that figure, when the authors pointed.
· Line 122: Phenotype should be Phenotypic
· Line 123: “transgenic lines had longer root than that wild type under mannitol treatment” please check for grammar.
· Line 126: “Based on the phenotype So the” So should be written with the small letter.
· Line 144: The authors have claimed that there are significant differences between treated lines and WT, but related phenotypes in left don’t show that.
· Line 149: The authors claimed that they used SD with “each with 100 seeds for each line”. I believe that these SD bars should not be as short as these.
· Line 158: Better to write utilization.
· Line 173: The asterisk are missed?
· Line 178: “necessary to example the ROS” what does mean?
· Line 182: “These results showed that WT leaves accumulated more ROS than that of transgenic lines under drought stress.” How the authors did get these results without any direct examination on ROS accumulation? So, the authors should perform qPCR using ROS1 gene to compare its relative before and after treatments or perform HPLC to show its contents before and after treatments to compare as a new figure.
· Line 189: “These results showed that overexpression of ZmASR3 reduced ROS accumulation by increasing the activities of SOD and CAT enzymes under drought condition.” the authors should perform qPCR using ROS1 gene to compare its relative before and after treatments or perform HPLC to show its contents before and after treatments to compare as a new figure.
· Line 198: Please confirm these analyses. It seems OE3-2 should be represented with three asterisk (p≤0.001).
· Line 208: One sample cannot be a sufficient criteria for comparing expression of relative genes. So, this figure cannot be used as a scientific figure. Please repeat these experiments using at least three independent plants with triplicate. Also, the authors should analyze these data using one comparing statistic software to compare and show significant differences.
· Figure S5: When the authors are telling about 100 seeds with triplicate experiments they should present standard deviation not standard errors.
· Figure S6: All data should be analyzed by statistical software to make comparison and show significant differences.
· Line 247: Please use half MS instead of “0.5 × MS medium” in through all manuscript.
· Figure 11: The authors should analyze the interaction of genes for assessing whether the difference between treatments more than expected or not (e.g. two way ANOVA).
· Some of cited references are old and should be up to date. For example, 1, 3, 4, 5, 10, 11, 12, 18, 20, 29, 32, 33, 36, 39, 40, 48, 49 and 50.
Author Response
Thank you for your comments concerning our manuscript. The comments are all valuable and very helpful for revising and improving our paper. In addition, they will also provide important guidance for our future research. We have studied the comments carefully and have made corrections accordingly to the new manuscript which we hope will meet with your approval. The main corrections in the marked versions of the revised manuscript and the responses to the reviewers’ comments are as follows.
Point 1: Please shorten the title to make easier understanding for all readers.
Response 1: New title “Functional analysis of the maize ASR gene family reveals ZmASR3, which positively regulates drought tolerance in transgenic Arabidopsis” can be found in the new manuscript.
Point 2: Line 13: “However, nothing is known about the ASR function in maize.” How did the authors think there is nothings about that, while I listed some of published papers about ASR in maize?
1. Bukan M;Sarcevic H;Buhinicek I;Palaversic B;Lewis RS, Kozumplik V. Stalk Rot Resistance in Maksimir 3 Synthetic Maize Population after Four Cycles of Recurrent Selection. Genetika-Belgrade. 2013,45(3),921-928, 10.2298/Gensr1303921b.
2. Cota LV;da Costa RV;Silva DD;Casela CR, Parreira DF. Quantification of Yield Losses Due to Anthracnose Stalk Rot on Corn in Brazilian Conditions. Journal of Phytopathology. 2012,160(11-12),680-684, 10.1111/jph.12008.
4. Li XY;Li LJ;Zuo SY;Li J, Wei S. Differentially expressed ZmASR genes associated with chilling tolerance in maize (Zea mays) varieties. Functional Plant Biology. 2018,45(12),1173-1180, 10.1071/FP17356.
5.Virlouvet L;Jacquemot MP;Gerentes D;Corti H;Bouton S;Gilard F;Valot B;Trouverie J;Tcherkez G;Falque M et al. The ZmASR1 Protein Influences Branched-Chain Amino Acid Biosynthesis and Maintains Kernel Yield in Maize under Water-Limited Conditions. Plant Physiology. 2011,157(2),917-936, 10.1104/pp.111.176818.
6. Zhang J;Zhu QS;Yu HJ;Li L;Zhang GQ;Chen X;Jiang MY, Tan M. Comprehensive Analysis of the Cadmium Tolerance of Abscisic Acid-, Stress- and Ripening-Induced Proteins (ASRs) in Maize. International Journal of Molecular Sciences. 2019,20(1), Artn 133. 10.3390/Ijms20010133.
7. Zhu JW;Huang XL;Liu T;Gao SG, Chen J. Cloning and function analysis of a drought-inducible gene associated with resistance to Curvularia leaf spot in maize. Molecular Biology Reports. 2012,39(8),7919-7926, 10.1007/s11033-012-1636-6.
Response 2: Sorry for this unscientific sentence. We revised the sentence in the new manuscript.
Point 3:
· Lines 64 and 65: “abiotic stress” should be abiotic stresses and also in across all the manuscript.
· Line 73: “but the exact function of maize ASR genes were still not well known”, what does it mean? Please explain clearly.
· Line 76: “member” change to members.
· Lines 66 and 74: ASR gene should be in italic and also across all the manuscript.
Response 3: Thank you for your suggestions, we modified these sentences in the new manuscript.
Point 4:
· Figure 2 caption: The authors have to analyze transcript levels to determine significant differences and show it using asterisk. Also which type of bars have been presented?
Response 4: Thank you for your suggestions. We analyzed the significant differences and showed it using asterisk. In addition, the SD bars had been presented in the new manuscript.
Point 5:
· Line 117: “the function of ZmASRs in responses to the abiotic stress still remain unclear in plants” please check for grammar.
· Line 120: The authors how did detected the homozygous lines? In biological processes sometimes may generate heterozygous embryos from homozygous and also reversely accidently.
· Supplemental figure 2: When the authors are talking about relative expression, they should analyze data via comparison statistical analyses and one control to exhibit significant differences between data. This process also has to be for supplemental figure 3. In addition, the authors have Witten SE, but according to the statistical analyses it should be Standard deviation. Moreover, I cannot find any asterisk in that figure, when the authors pointed.
· Line 122: Phenotype should be Phenotypic
· Line 123: “transgenic lines had longer root than that wild type under mannitol treatment” please check for grammar.
· Line 126: “Based on the phenotype So the” So should be written with the small letter.
Response 5:
Line 117: we modified the sentence in the new manuscript.
Line 120: In general, if all of seeds of independent T3 line can be germinated on MS medium containing 25 μg mL−1 hygromycin, so this line possibly be homozygous line.
Supplemental figure 2: we modified these places in the new manuscript. It is not necessary to analyze the significant differences of relative expression via comparison statistical analyses as these selected lines are homozygous that have different expression level from many transgenic lines. In addition, the asterisk can be found in the Supplemental figure 3-1.
Line 122,123 and 126: we revised these sentences in the new manuscript.
Point 6:
· Line 144: The authors have claimed that there are significant differences between treated lines and WT, but related phenotypes in left don’t show that.
Response 6: I don’t know if you are talking about fig. 4.The phenotypes in fig.4 is noticeable.
Point 7:
· Line 149: The authors claimed that they used SD with “each with 100 seeds for each line”. I believe that these SD bars should not be as short as these.
· Line 158: Better to write utilization.
· Line 173: The asterisk are missed?
Response 7: Thank you for your warning. We revised these places in the new manuscript.
Point 8:
· Line 178: “necessary to example the ROS” what does mean?
· Line 182: “These results showed that WT leaves accumulated more ROS than that of transgenic lines under drought stress.” How the authors did get these results without any direct examination on ROS accumulation? So, the authors should perform qPCR using ROS1 gene to compare its relative before and after treatments or perform HPLC to show its contents before and after treatments to compare as a new figure.
· Line 189: “These results showed that overexpression of ZmASR3 reduced ROS accumulation by increasing the activities of SOD and CAT enzymes under drought condition.” the authors should perform qPCR using ROS1 gene to compare its relative before and after treatments or perform HPLC to show its contents before and after treatments to compare as a new figure.
Response 8:
Line 178: Sorry for the default sentences. The right sentence“necessary to examine the ROS”was include in the new manuscript.
Line 182 and 189: In fact, H2O2 is the main type of ROS. Our results showed that lower H2O2 content in transgenic line than WT by the DAB staining assay. So we can get these results. In addition, the expression level of AtCAT3 and AtSOD1 genes, which are involved in ROS detoxification, were were up-regulated in the ZmASR3 overexpression transgenic lines under the drought treatment. Therefore, these results can support my conclusion overall.
Point 9:
· Line 198: Please confirm these analyses. It seems OE3-2 should be represented with three asterisk (p≤0.001).
· Line 208: One sample cannot be a sufficient criteria for comparing expression of relative genes. So, this figure cannot be used as a scientific figure. Please repeat these experiments using at least three independent plants with triplicate. Also, the authors should analyze these data using one comparing statistic software to compare and show significant differences.
· Figure S5: When the authors are telling about 100 seeds with triplicate experiments they should present standard deviation not standard errors.
· Figure S6: All data should be analyzed by statistical software to make comparison and show significant differences.
· Line 247: Please use half MS instead of “0.5 × MS medium” in through all manuscript.
· Figure 11: The authors should analyze the interaction of genes for assessing whether the difference between treatments more than expected or not (e.g. two way ANOVA).
Response 9:
Line 198: We checked and corrected it, thank you!
Line 208: The experimental sample is a mixture of three genetically modified lines. The results are averaged over the three lines and will therefore be more accurate, with relatively smaller errors that are more in line with experimental specifications and statistical significance.
Figure S5: We modified this in the new manuscript.
Figure S6: No significantly difference was found in each treatment.
Line 247: We modified these places in the new manuscript.
Figure 11: We analyzed the interaction of genes according to your suggestion, thank you for your professional suggestions.
Point 10:
· Some of cited references are old and should be up to date. For example, 1, 3, 4, 5, 10, 11, 12, 18, 20, 29, 32, 33, 36, 39, 40, 48, 49 and 50
Response 10: Thank you for your suggestions, we replaced these old references with new references in the revised manuscript.
Round 2
Reviewer 1 Report
Compared to the previous version, the main change is that statistical data and significance have been added to figures. But results have not really been further discussed. Over-expressing lines are more resistant to exogenous ABA (better germination, longer roots when ABA is added to the medium), but also produce more ABA and are more resistant to drought, suggesting that the sensitivity to exogenous ABA is not directly correlated to drought resistance. This phenotype might be compared to that of other ABA-relative mutants such as ABI1-1, which has been deeply characterized. Like ZMASR3 OE plants, ABI1-1 is ABA-insensitive at the germination stage, has higher ABA contents, but it has also a high leaf transpiration rate and low proline content. To connect ZmASR3 to ABA-related pathways, is it possible to find (or not) a mutant that looks like over-expressing lines? And what might be the role of ZmASR3 and other ASR proteins in the nucleus? Direct or indirect regulation of transcription factors?
Also, the drafting still needs to be improved all along the manuscript.
Author Response
Point 1: Compared to the previous version, the main change is that statistical data and significance have been added to figures. But results have not really been further discussed. Over-expressing lines are more resistant to exogenous ABA (better germination, longer roots when ABA is added to the medium), but also produce more ABA and are more resistant to drought, suggesting that the sensitivity to exogenous ABA is not directly correlated to drought resistance. This phenotype might be compared to that of other ABA-relative mutants such as ABI1-1, which has been deeply characterized. Like ZMASR3 OE plants, ABI1-1 is ABA-insensitive at the germination stage, has higher ABA contents, but it has also a high leaf transpiration rate and low proline content. To connect ZmASR3 to ABA-related pathways, is it possible to find (or not) a mutant that looks like over-expressing lines? And what might be the role of ZmASR3 and other ASR proteins in the nucleus? Direct or indirect regulation of transcription factors?
Also, the drafting still needs to be improved all along the manuscript.
Response 1: Thank you for your kind comments and professional suggestions. Results on the ABA sensitivity and ZmASRs localization were further discussed and included in the marked versions of the revised manuscript which we hope will meet with your approval. In addition, the drafting was further modified. Thank you again!
Reviewer 2 Report
1- Title: One suggestion for authors: “ZmASR3 gene from maize ASR gene family positively regulates drought tolerance in transgenic Arabidopsis”
2- Line 4: Arabidopsis shouldn’t be italic.
3- Line 13: “However, little is known about the function in enhancing…” is not clear. Please make one readable and scientific.
4- Line 98: “Significant levels” should be significant differences. Also across whole manuscript.
5- Figure 2: Which parametric or nonparametric statistical analyzing has been used to analyze these data? If assume that the authors are calculating n≥3 (not mentioned) plants for each columns, please clarify why did not reveal standard errors (SE)? This problem also will be happened for figures 4, 5, 6, 7.
6- The authors should add one separate statistical analyses part in Material and Methods to explain how they analyzed their data completely.
7- Line 149: I cannot find * P < 0.05; ***P < 0.001.
8- Line 156: The same problem with No. 7
9- Line 172: “each with 200 stomata for each line”, please rewrite.
10- Line 198: “* P < 0.05” ???
11- Figure 8: How many biological repeats? How many replicates? “* P < 0.05” ???
12- Figure 9: The authors only presented replicates. How many biological repeats? “*P < 0.05; ***P < 0.001.” ???
13- Line 253: “***P < 0.001” ??
14- Figure 11: How do the authors mention asterisks while there is no one in figure?
15- According to the authors’ answer (Response 9), they analyzed the interaction of genes. Unfortunately I could not find that’s explanation through manuscript.
Author Response
Point 1: Title: One suggestion for authors: “ZmASR3 gene from maize ASR gene family positively regulates drought tolerance in transgenic Arabidopsis”
Response 1: According to your professional suggestion, we modified the title of manuscript. Thank you.
Point 2:
2- Line 4: Arabidopsis shouldn’t be italic.
3- Line 13: “However, little is known about the function in enhancing…” is not clear. Please make one readable and scientific.
4- Line 98: “Significant levels” should be significant differences. Also across whole manuscript.
Response 2: We modified these places in the revised manuscript. Thank you.
Point 3:
5- Figure 2: Which parametric or nonparametric statistical analyzing has been used to analyze these data? If assume that the authors are calculating n≥3 (not mentioned) plants for each columns, please clarify why did not reveal standard errors (SE)? This problem also will be happened for figures 4, 5, 6, 7.
6- The authors should add one separate statistical analyses part in Material and Methods to explain how they analyzed their data completely.
7- Line 149: I cannot find * P < 0.05; ***P < 0.001.
8- Line 156: The same problem with No. 7.
9- Line 172: “each with 200 stomata for each line”, please rewrite.
10- Line 198: “* P < 0.05” ???
11- Figure 8: How many biological repeats? How many replicates? “* P < 0.05” ???
12- Figure 9: The authors only presented replicates. How many biological repeats? “*P < 0.05; ***P < 0.001.” ???
13- Line 253: “***P < 0.001” ??
14- Figure 11: How do the authors mention asterisks while there is no one in figure?
Response 3: Thank you for your kind comments and professional suggestions. We read the related references and consulted some professional teachers on the biostatistical method. One separate statistical analyses part was added in the Material and Methods section and some false places were corrected in the revised manuscript based on our understanding. I hope that these corrections are reasonable and will meet with your approval. Thank you again.
Point 4:
15- According to the authors’ answer (Response 9), they analyzed the interaction of genes. Unfortunately I could not find that’s explanation through manuscript.
Response 4: Thank you for your comments. We did not understand rightly your meanings of the round 1 suggestions on this question. In fact, as we all known, these genes were involved in different pathways, so it is not necessary analyze the interaction of genes in our experiments. Thank you again.
Round 3
Reviewer 1 Report
New elements have been brought to the discussion, which improves the paper. There are still problems with the wording of some sentences (English editing required). Also,
the title of figure 11 seems to be inappropriate ("Tu of ABA biosynthesis", not only ZmASR3 transcript, not only mannitol treatment), and meaning of the sentence lines 325-326 is unclear (I would rather say that ZmASR3 uses an ABA signaling pathway that probably does not require ABI1-like PP2Cs).
Author Response
Point 1: New elements have been brought to the discussion, which improves the paper. There are still problems with the wording of some sentences (English editing required). Also, the title of figure 11 seems to be inappropriate ("Tu of ABA biosynthesis", not only ZmASR3 transcript, not only mannitol treatment), and meaning of the sentence lines 325-326 is unclear (I would rather say that ZmASR3 uses an ABA signaling pathway that probably does not require ABI1-like PP2Cs).
Response 1: We rewrote the manuscript and enlisted an English language expert to carefully check the grammar and syntax. In addition, according to your kind comments, the title of fig 11 and the sentence lines 325-326 were modified in the new manuscript.
Reviewer 2 Report
Please find the comments through the attached PDF.

Author Response
Point 1: Line 2: ZmASR3 shouldn be italic. Line157、Line164、Line181、Line206、Line217、Line234、Line261、Line266、Line411: they should be ≥ not Line266-267: Better write: ANOVA has been used to reveal significant differences.
Response 1: Thank you for your kind comments. According to your suggestions, these palces were modified in the revised manuscript.